# Pilot and Feasibility of Combining a Medication Adherence Intervention and Group Diabetes Education for Patients with Type-2 Diabetes

**DOI:** 10.3390/pharmacy7030076

**Published:** 2019-06-28

**Authors:** Matthew Witry, Melissa Ernzen, Anthony Pape, Brahmendra Reddy Viyyuri

**Affiliations:** 1Department of Pharmacy Practice and Science, University of Iowa College of Pharmacy, Iowa City, IA 52242, USA; 2Mercy Family Pharmacy, Dubuque, IA 52001, USA

**Keywords:** diabetes education, pharmacist, medication synchronization, medication adherence, diabetes self-management, patient-reported outcomes

## Abstract

**Introduction**: Controlling diabetes typically requires self-management and medications. Community pharmacists are positioned to support patients with both. **Methods**: This study assessed the feasibility and potential benefit of combining pharmacist-provided group diabetes education (up to eight sessions) and medication synchronization using a three-group design. Data were collected using pre–post paper surveys and electronic health record data. One group received both education and synchronization services, another group received medication synchronization only, and a third served as control. **Results**: Of 300 contacted patients, eighteen patients participated in group diabetes education, 14 had medication synchronization only, and 12 comprised a control group. There was little change in HbA1c over the study period. Medication adherence appeared to be positively aided by medication synchronization, although all groups started with high adherence. Some medication beliefs and self-care activities may have been positively impacted by group diabetes education. Both groups receiving medication synchronization were satisfied. **Conclusions**: Participants strongly agreed they would recommend group diabetes education from the study pharmacy to a friend and were satisfied with medication synchronization; however, it was difficult to tell if there was a synergistic effect by combining the two services. Reimbursement for diabetes education was not obtained despite multiple attempts, hindering sustainability.

## 1. Introduction

As of 2015, an estimated 23.1 million Americans were diagnosed with type 2 diabetes mellitus (T2D), with another 7.2 million having undiagnosed diabetes [1]. Treating diabetes and its complications is estimated to cost the U.S. healthcare system $327 billion annually [1,2]. Approximately 91.4% of the economic burden of diabetes is attributed to T2D as its prevalence is significantly higher than T1D [3]. While prevention of T2D is a vital public health issue, for those with T2D increased medication adherence has been associated with better glycemic control [4] and decreased risk of hospitalization and emergency department use [5]. Medication adherence is important for patients with T2D as they often have comorbidities such as hypertension, dyslipidemia, coronary artery disease, and depression—conditions commonly controlled with medication therapy.

Medication nonadherence is a significant barrier to the chronic management of patients with T2D [6]. Rates of nonadherence for oral anti-diabetic medications range from 36%–93% [7]. Nonadherence is multifactorial and includes forgetfulness, complex multi-dose regimens, medication organization, concerns about side effects, cost, low feelings of medication need, health literacy and numeracy deficits, and stigma associated with diabetes and chronic medication use [7,8,9,10,11,12,13]. Studies of adherence interventions, including by pharmacists, have found that multifaceted approaches produce the most robust improvements in outcomes [14,15,16]. These include the pairing of medication organization interventions and disease-state education [14,15,16]. 

Self-management of diabetes is also paramount to preventing complications [17,18]. Behaviors like diet, exercise, and blood glucose monitoring improve outcomes, but can be challenging to maintain [6]. Diabetes education by certified educators has been found useful for improving outcomes [19,20], including when provided by pharmacists [21]. Difficulties with obtaining the needed experience hours and limited financial returns have contributed to only a minority of pharmacists becoming certified diabetes educators [22,23]. Community-based pharmacists are uniquely positioned to engage with patients with T2D because of interactions during medication dispensing. Engaging patients in the medication dispensing process provides an opportunity for additional education and reinforcement before and after they begin taking the medication on their own [24]. Refill counseling, however, is often less emphasized than new prescription counseling [25,26], and therefore targeted interventions are often used [27].

An emerging service provided by community pharmacies to support medication adherence is to consolidate medication refills to the same date in a process often called medication synchronization [28,29,30]. When implemented with an accompanying clinical intervention by the pharmacist, this is called an appointment-based model (ABM) [29,30]. Medication synchronization programs have had a positive impact on medication adherence [31,32], and patients and pharmacists have expressed positive attitudes toward these programs [33,34]. 

The objective of this study was to assess the feasibility of a combined medication synchronization and diabetes education program. Measures included patient-reported satisfaction, beliefs about medicines, and diabetes self-management activities. Comparisons were made with a control group and a medication synchronization only group to identify potential areas of strength and weakness.

## 2. Materials and Methods 

This feasibility study was conducted at a single location of a small independent community pharmacy chain in the Midwest U.S. The pharmacy has a history of providing clinically oriented services such as medication therapy management. The study was approved by the Institutional Review Board (IRB) of the health system, of which the pharmacy is affiliated. Approval was granted via a signed letter from the chairperson of the Joint Quality Oversight Committee of the IRB to M.E. on 14 June 2017 for 1 year. Participants were given written details about the study and provided written consent to participate.

### 2.1. Participants

The sampling frame was comprised of patients aged 18 years and older who have used the study pharmacy to fill at least one anti-diabetic medication within the previous 60 days and take two or more other chronic oral prescription medications. There were no additional inclusion criteria based on historical medication non-adherence. 

### 2.2. Group Assignment and Recruitment

The pharmacy used a 3rd party web-based quality improvement and documentation platform to select the population of patients that met the above inclusion criteria. This involved creating a custom query that selected patients that were 18 years or older, fill an oral anti-diabetic medication (e.g., biguanides, meglitinides, DPP-4 inhibitors, SGLT2 inhibitors, thiazolidinediones), and take two or more additional oral prescription medications. From this list, 300 patients were selected at random using values assigned to each entry using a random number generator. The justification for 300 participants was that we expected 7–10% participation, which would translate to about 12–18 patients participating in group diabetes education. Of the 300 patients in the sample, a random one hundred patients received a mailing for the control condition inviting them to complete a survey to help the pharmacy better understand the patients they serve. Two hundred patients received a mailing offering participation in group diabetes education and an adherence program. A brief telephone call by one of the study pharmacists was used to inquire about participation and answer questions.

### 2.3. Initial Assessment

All patients receiving the mailings were instructed to complete the enclosed survey, which assessed inclusion criteria and baseline adherence attitudes, and return it to the pharmacy for a $15 gift card either to the pharmacy or to the local grocery store. 

Patients returning the survey in exchange for the gift card went through a brief enrollment session in the pharmacy. If the patient was interested in diabetes education and could accommodate it with their schedule, they could enroll in the diabetes education arm of the pilot. Patients that were not interested in diabetes education or could not fit the specified times into their schedule could enroll in the medication synchronization service only arm. 

### 2.4. Diabetes Education and Medication Synchronization Arm

Patients enrolling in the diabetes education arm were scheduled to meet with the certified diabetes educator (CDE) pharmacist for an initial visit. During this appointment, the pharmacist performed an initial assessment and received permission from the patient to request an order for group diabetes education from his or her primary care physician or advanced-level prescriber. 

Next, patients attended up to eight, 1 h group diabetes education classes held at a community room on the local hospital campus. The CDE pharmacist facilitated the classes. The classes included an overview of diabetes, education on short- and long-term complications, nutrition and exercise, and other topics related to their diabetes diagnosis. The patients could choose if they wanted to attend the class over the lunch hour or in the early evening. 

After completion of the diabetes education classes, patients met with the pharmacist to be assessed for medication adherence barriers and their interest in medication synchronization. The medication synchronization process was enhanced for this group, in that it contained a set of education discussion points to reinforce the diabetes education topics. The CDE pharmacist performed the calls. Three topics were addressed—meal planning and carb counting, blood glucose monitoring, and monitoring long-term complications. These one-on-one discussions were designed to range from 5–10 min depending on patient questions and discussion.

### 2.5. Medication Synchronization Service Only Arm

Patients in the medication synchronization only arm were scheduled for an appointment to meet with the study pharmacist to discuss medication adherence barriers and their fitness for having their medications synchronized. If medication synchronization was implemented, the patient received a follow-up phone call approximately three days before medications were due to be filled. This call served to assess emerging adherence barriers, especially if the patient claimed he or she did not need a particular medication refilled at the time. The call also allowed for the patient to ask questions of the pharmacist or the pharmacist to ask questions of the patient. The medications were then refilled for the agreed upon appointment date.

### 2.6. Control Arm

Patients returning the survey for the control condition were not offered diabetes education or the medication synchronization service. This group was incentivized to complete a baseline and end-of-study survey to compare with the other groups.

### 2.7. Measurements and Variables

Demographics collected included age, gender, insurance, and type of oral diabetes medications. Patients also completed the 10-item Beliefs about Medicines Questionnaire (BMQ) [35] and a 10-item measure of diabetes self-management activities [36]. Change in medication adherence was measured using a proportion of days covered (PDC) statistic for oral anti-diabetic medications (e.g., metformin, glipizide). PDC changes were assessed using a 6 month pre-randomization period and the 6 months following the receipt of the first intervention. For control patients, the mean time to intervention was substituted for their pre–post intervention gap. 

Patient satisfaction was measured for the group diabetes education classes and the medication synchronization service using survey items developed for this study. Diabetes education was assessed using 18 items developed for the study. Medication synchronization was evaluated using eight items developed for the study. Two additional items evaluated the diabetes education component of the medication synchronization service for the intervention group. While there was an existing scale for medication synchronization satisfaction [34], it used an overall affective orientation, whereas the present feasibility study used a performance orientation [37].

These variables were chosen to allow the team to assess the feasibility [38] of the two services related to recruitment approach, patient interest, targeting criteria, and patient satisfaction. We also sought to understand the diabetes self-care behaviors, medication beliefs, medication adherence, and HbA1c of those agreeing to participate. 

### 2.8. Data Collection

The above variables were collected using paper surveys, which were limited to the front and back of one page to decrease the burden on patients. Surveys for the interventions were handed out following the interventions, and patients could either complete them on site or take them home to finish. Postage paid return envelopes were used to increase response. Study pharmacists also called patients to remind them to complete study materials if needed.

### 2.9. Data Analysis

Data management was performed using Microsoft Excel (Redmond WA), and analyses were conducted using R v.3.4. Descriptive statistics were used for items. Formal tests for differences were not performed given the goal was to investigate feasibility rather than the effectiveness of the intervention. Internal consistency of the BMQ and patient satisfaction survey items were assessed using Cronbach’s alpha test.

## 3. Results

Of the 300 invitation letters sent to patients taking oral anti-diabetic medications, 18 persons completed group diabetes education and the adherence intervention, 13 participated in medication synchronization only, and 12 comprised a control group. Thus, 9% of those offered group diabetes education were both interested in the intervention and could fit one of the two times into their schedules. Of the remaining patients targeted for the intervention, 7.1% agreed to engage with the medication synchronization alternative intervention. Thirteen patients (13%) completed surveys as part of a comparison group. 

Patient demographics are detailed in Table 1. The ages in the diabetes education group skewed younger, and they had diabetes for fewer years than the other groups. The patients participating in diabetes education also took more medications on average.

The pharmacy utilized their electronic medical record to access HbA1c, which yielded two values for all consenting participants during the study period. Changes were similar across the three groups (Figure 1). There appeared to be a benefit to PDCs associated with medication synchronization, although most patients remained above the 80% threshold for being considered adherent (Figure 2).

In the diabetes education group, the intervention led to increased necessity beliefs (Figure 3) and decreased concern beliefs (Figure 4). The control group also saw a slight decrease in concern beliefs (Figure 5), but their necessity beliefs were also decreased (Figure 6). Patients in the diabetes education group appeared to have positive changes in their self-management activities (Figure 7). The control group already had high rates of performing the activities (Appendix A). Both the diabetes education and control groups reported high levels of medication taking (6 versus 7 of last 7 days on average, respectively), and both groups reported checking their blood glucose more often at the end of the study period. One subject in the diabetes education group reported a substantial decline in their self-management activities over the study period, although no reason could be determined after rechecking the data. 

Patients generally were satisfied with the group diabetes education classes provided by the CDE pharmacist (Table 2). For most items, respondents agreed or strongly agreed with satisfaction statements regarding the delivery and structure of the program. Patients reported they would recommend the class and that the classes were worth the time invested. Only 1 of 13 reported the course did not motivate them to stay in control of their diabetes. 

Participants also were satisfied with the medication synchronization intervention and would recommend the program (Figure 8). Two participants, however, disagreed that the program fit with their budget. For those in the diabetes education group, they responded positively about the short follow-up discussions reinforcing diabetes education topics from class during the first medication synchronization calls.

The pharmacy sought reimbursement for providing patients with group diabetes education from Medicare (Medicaid managed care and private insurance plans). No claims were successfully paid. Rejections were based on insurers offering diabetes support by their staff and insurers requiring additional costly pharmacy-level certifications.

## 4. Discussion

Overall, patients were satisfied with the group diabetes education provided by the CDE pharmacist and the medication synchronization service. The participation rate, however, was fairly low, with only 7% of those offered completing the full program. Others have reported that patients have limited knowledge of specialized diabetes services provided at community pharmacies, including those provided by CDE pharmacists [39]. The study pharmacists were successfully able to get prescriptions from each patient’s primary care provider for group diabetes education without resistance. The prescribers may have had some awareness of the study pharmacist and pharmacy as providers of progressive clinical pharmacy services.

For those receiving both study interventions, it was difficult, however, to determine if there was a combined benefit for pairing medication synchronization with group diabetes education that exceeded either activity alone. All of the patients participating in the group diabetes education subsequently signed up to have their medications synchronized, which may suggest a possible priming effect. Alternatively, the patients that were willing to sign up for group diabetes education may have an overall positive view of pharmacy services, contributing to their signing up for both. 

While this feasibility study was not designed to statistically assess the effectiveness of the intervention to improve clinical measures like HbA1c and PDC, some patients in the sample did appear to experience positive changes. Medication synchronization has been linked to patients refilling their prescriptions more regularly [31,40], although the link to clinical outcomes has yet to be established. Studies of pharmacist-led diabetes interventions are mixed, particularly with their ability to decrease HbA1c through interventions focused on education and self-care counseling [27,41]. The study period likely was too short to measure the change in HbA1c. Also, it likely would require a large sample to show HbA1c improvement with established patients who already are under reasonably good control.

Moving on to medication beliefs, on average, participant responses to BMQ items related to necessity and concerns did not change dramatically. Part of this may be that the study did not select for patients with low adherence at baseline. Doing so likely would increase the potential impact of the intervention. Again, these patients generally were in the maintenance stage of their treatments rather than patients new to therapy. As a result, issues related to side effects, negative beliefs, or other factors may already have led to changes in therapy or those patients discontinuing their treatment. Also, the use of a convenience sample may have selected for patients with more positive medication beliefs than in the general population. 

The relatively small proportion of patients having negative beliefs about their medicines, low adherence, or sub-optimal HbA1c values meant these patients had little room for improvement. This model likely would have benefited from more strict screening and targeting of patients to include those with more obvious need and who are at higher risk for problems. In the future, focusing on patients with the greatest need likely would have a greater impact [16].

In addition to focusing on patients with a more pronounced need, recent studies using community pharmacists to improve adherence to diabetes medications have used structured interview guides, such as the Drug Adherence Workup (DRAW) to more comprehensively assess patient barriers to adherence [42,43,44]. Consistently using open-ended questions also may help improve adherence for patients with diabetes [45]. An interview guide was not used in the present study since the diabetes education arm received the additional education reinforcement during their medication synchronization calls. Pharmacists may need additional training to use such approaches to share decision-making in the community pharmacy setting [46].

Since most of these patients had participated in formal diabetes education before, pharmacists should consider what an established patient may need to manage their diabetes. While patients may need reinforcement of their self-care, there is evidence that patients with T2D may prefer medication-focused content and recommendations from the pharmacist [47]. Patients may also be more comfortable when the pharmacist has a collaborative relationship with the prescriber [47]. The pharmacist in this study received a prescription for all the participants to receive group diabetes education from each patient’s prescriber, which may have made patients feel more comfortable participating.

Another potential area that relates to both diabetes education and adherence to diabetes education is including a focus on reducing diabetes distress. Pharmacists should be cognizant of the burden that self-care behaviors and taking multiple chronic medications can place on a patient with diabetes. A study by Machen et al. demonstrated that community pharmacists could decrease diabetes distress among participants [48]. Measures of diabetes distress also could be used to screen patients.

Perhaps the most significant barrier to feasibility encountered was not being reimbursed by insurers for any of the group diabetes education claims [49]. This is despite the pharmacist having a CDE credential, program certification with the public health department, and multiple claim resubmissions. Alternative sustainability options include partnering with local non-profit organizations that help the underserved, partnering with employers, new reimbursement options like chronic care management (CCM), and annual wellness visits. Looking forward, the potential passage of provider status legislation whereby pharmacists would have increased authority to bill independently for certain services may open up opportunities in the U.S., although individual plans may still choose to offer the service in-house or with a restricted network of providers. 

Limitations of this feasibility study include potential selection bias, which may have led to the sickest patients not being targeted. Also, while there was a random component to group assignment, the medication synchronization only arm was the patients not willing to participate in group diabetes education, which likely led to a selection bias. Another limitation to the adherence analysis was several intervention and control patients having their oral anti-diabetic medications discontinued. This occurred because diabetes control improved or because insulin was initiated instead. In one case, the patient’s dose was decreased without a new prescription being written, so they appeared nonadherent. These patient-specific situations make comparing PDC differences challenging, potentially misleading, and therefore should be considered carefully. Lastly, there is a possibility that participants in the intervention groups behaved differently because they knew they were in a study and being monitored. Patients in the control group likely would not be subject to a potential Hawthorne effect.

## 5. Conclusions

The study pharmacists found it feasible to identify a small group of interested patients, obtain a prescription for diabetes education from their prescribers, and deliver the group sessions for which most were satisfied. Patients were similarly satisfied with a medication synchronization intervention, although it was difficult to tell if there was a synergistic effect between the two services. Pharmacists looking to provide diabetes education should carefully investigate reimbursement options, including nontraditional arrangements as the CDE pharmacist in the present study was not able to receive reimbursement directly from insurers.

## Figures and Tables

**Figure 1 pharmacy-07-00076-f001:**
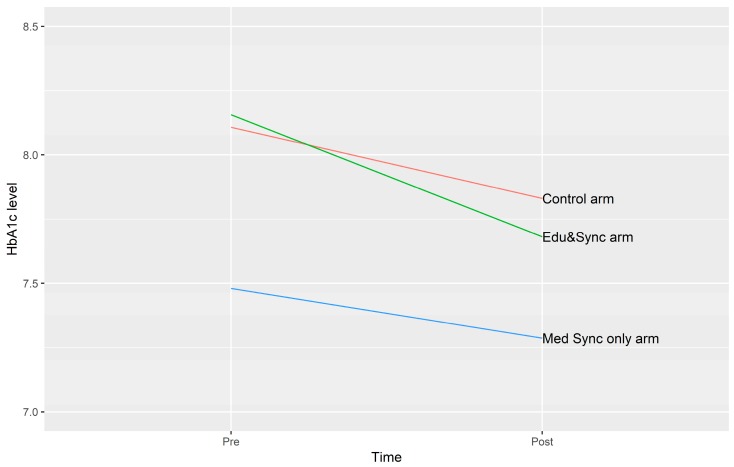
HbA1c differences in all three groups.

**Figure 2 pharmacy-07-00076-f002:**
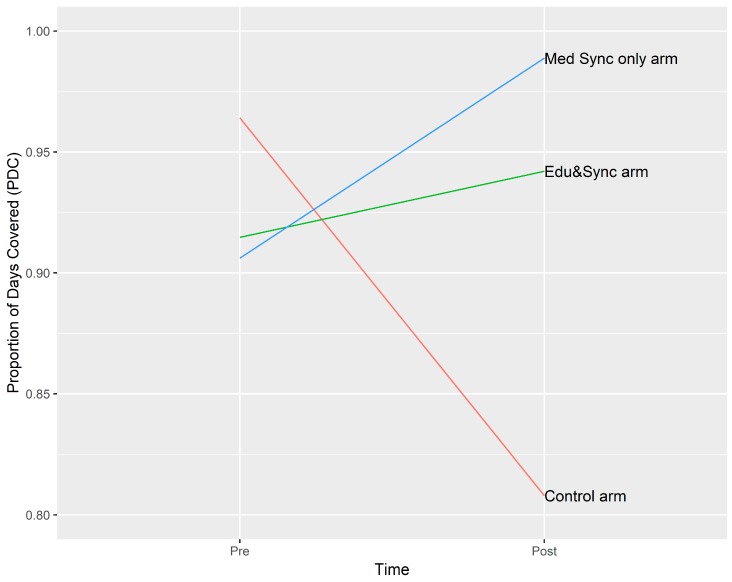
PDC (proportion of days covered) count differences in all three groups.

**Figure 3 pharmacy-07-00076-f003:**
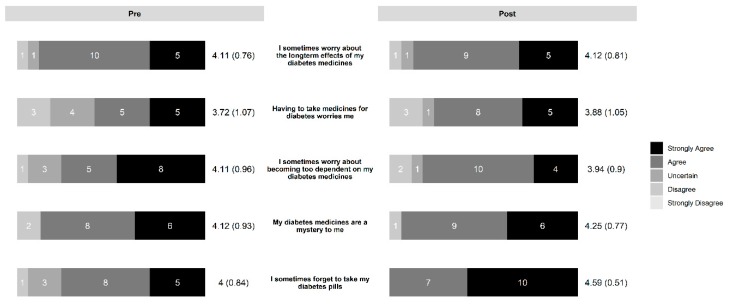
Change in necessity beliefs in diabetes education group.

**Figure 4 pharmacy-07-00076-f004:**
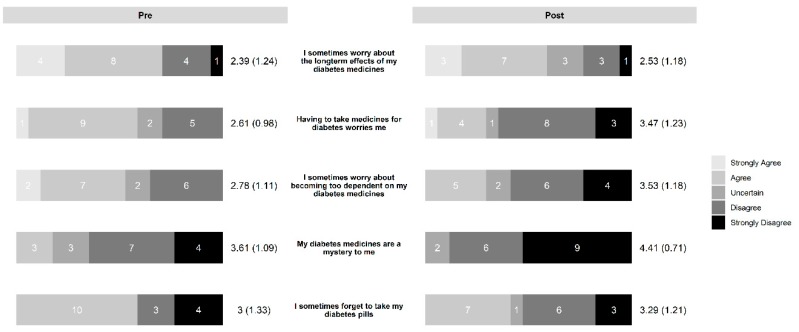
Change in concern beliefs in diabetes education group.

**Figure 5 pharmacy-07-00076-f005:**
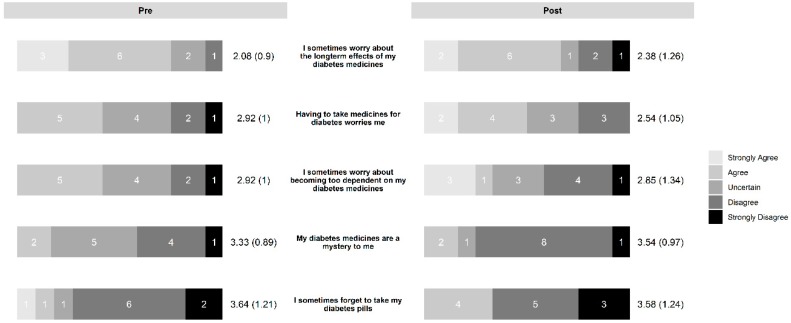
Change in concern beliefs in control group.

**Figure 6 pharmacy-07-00076-f006:**
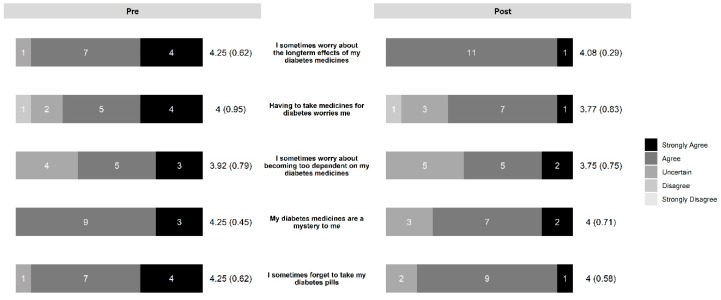
Change in necessity beliefs in control group.

**Figure 7 pharmacy-07-00076-f007:**
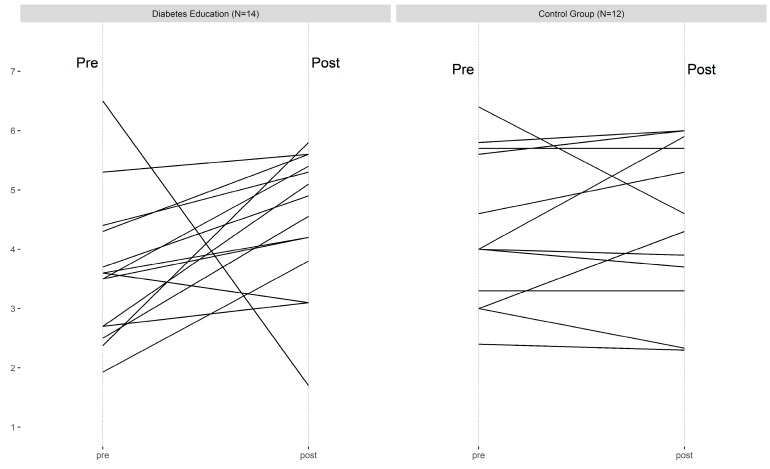
Diabetes self-management activities for diabetes education and control groups.

**Figure 8 pharmacy-07-00076-f008:**
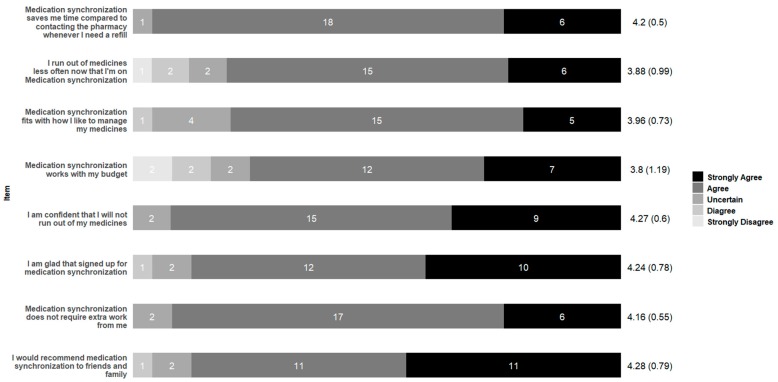
Patient satisfaction with a medication synchronization service.

**Table 1 pharmacy-07-00076-t001:** Participant Demographics.

Characteristic	Education and Medication Synchronization Arm	Medication Synchronization Service Only Arm	Control Arm
Gender			
Male	7 (38.89%)	6 (40%)	5 (38.5%)
Female	11 (61%)	7 (46.6%)	8 (61.5%)
Age			
18–29	1 (5.5%)		
30–39	2 (11.1%)	1 (6.6%)	
40–49	3 (16.6%)		1 (7.6%)
50–59	6 (33.3%)	4 (26.6%)	3 (23%)
60–69	5 (27.7%)	8 (53.3%)	7 (53.8%)
70 or older	1 (5.5%)	2 (13.3%)	2 (15.3%)
Education			
Grade school	1 (5.5%)	2 (13.3%)	1 (7.6%)
High school	7 (38.8%)	6 (40%)	5 (38.4%)
Technical/2-yr	3 (16.6%)	3 (20%)	2 (15.3%)
Some college	2 (11.1%)		4 (30.7%)
College Graduate	2 (11.1%)	4 (26.6%)	
Advanced degree	1 (5.5%)		1 (7.6%)
Insurance			
None		1 (6.6%)	
Medicaid	4 (22.2%)	9 (60%)	6 (46.15%)
Medicare	6 (33.3%)	5 (33.3%)	4 (30.77%)
Plan through work	7 (38.8%)	2 (13.3%)	2 (15.4%)
Individual plan	3 (16.6%)	5 (33.3%)	2 (15.4%)
Currently live with?			
No one	5 (27.78%)	6 (40%)	4 (30%)
Spouse or Partner	9 (50%)	7 (46.6%)	6 (46.1%)
Dependent Children	1 (5.56%)		2 (15.3%)
Adult Children	1 (5.56%)		2 (15.3%)
Friend or Room mate	2 (11.11%)	1 (6.6%)	1 (7.7%)
Altogether, how many prescription medicines do you take every day? (Not just for diabetes) (mean (SD))	8.07 (2.81)	7.33 (3.37)	9.00 (4.44)
How would you rate your overall health? ^a^ (mean (SD))	2.86 (0.68)	2.67 (0.72)	3.00 (0.71)
How often do you need to have someone help you when you read instructions, pamphlets, or other written materials from your doctor of pharmacy? ^b^ (mean (SD))	3.88 (1.52)	3.6 (1.35)	4.38 (0.96)
Do you use an insulin injection?			
Yes	5 (27.7%)	5 (33.3%)	6 (46.1%)
No	13 (72.3%)	10 (66.6%)	7 (53.9%)
How long have you had your Type 2 diabetes? (Years) (mean (SD))	6.62 (5.20)	7.47 (5.84)	8.08 (6.62)

^a^ 1 Poor, 2 Fair, 3 Good, 4 Very Good, 5 Excellent; ^b^ 1 Never, 2 Rarely, 3 Sometimes, 4 Often, 5 Always.

**Table 2 pharmacy-07-00076-t002:** Patient satisfaction with diabetes education classes (n = 13) ^a^.

Item	Agreement Mean (SD)
The time for the diabetes education classes was convenient.	4.38 (0.51)
Traveling to the diabetes education classes was easy.	4.38 (0.51)
The pharmacist talked in a way that was easy to understand.	4.77 (0.44)
Talking to other people with diabetes was helpful.	4.54 (0.66)
The pharmacist was good about answering my questions.	4.69 (0.48)
The pharmacist made good suggestions about how to eat healthier.	4.69 (0.48)
The pharmacist made good suggestions about how to fit exercise into my day.	4.46 (0.52)
I am delighted with the diabetes education classes.	4.69 (0.48)
I am glad that I signed up to take these diabetes education classes.	4.69 (0.48)
Coming to the diabetes education classes was worth my time.	4.62 (0.51)
What I got from the diabetes education classes was worth me showing up.	4.54 (0.52)
I am more confident that l can manage my diabetes sense taking these classes.	4.38 (0.65)
The quality of the diabetes education classes met my standards.	4.54 (0.52)
The quality of the diabetes education classes exceeded my expectations.	4.31 (0.85)
I am confident that I were have better control of my blood sugars because of these classes.	4.38 (0.65)
I am confident that can keep up with my diabetes management long-term.	4.31 (0.63)
The classes motivated me to stay in control of my diabetes.	4.31 (0.95)
I would recommend group diabetes education from this pharmacy to my friends and family.	4.77 (0.44)

^a^ 1 Strongly Disagree, 2 Disagree, 3 Uncertain, 4 Agree, 5 Strongly Agree.

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
