# Peer review of "Pilot and Feasibility of Combining a Medication Adherence Intervention and Group Diabetes Education for Patients with Type-2 Diabetes"

_pharmacy, 2019, doi:10.3390/pharmacy7030076_

Round 1
Reviewer 1 Report
This feasibility study will be of limited interest to readers not withstanding that it is generally conducted in a scientific rigorous method. The limited interest is mainly attributable to the poor statistical power of the study (low participation rate) and limitations of the study design (e.g. single center-Mercy Family Pharmacy) which makes it impossible to generalize from the findings. There are some aspects of the feasibility of such a service that are not measured which it might be useful to declare. It would have been useful if a conceptual framework (e.g. UK MRC Framework for complex interventions) was contemplated in the paper.
Author Response
Please see the attached response. Thank you for taking the time and care to review this manuscript.

Reviewer 2 Report
The authors are commended for designing and undertaking this study. Implementation is important and we want to see and learn from the results independent of the size of the effects observed.
The manuscript can probably be improved if the methods are described in some more detail, if certain statistics are calculated and if the implementation-hurdles are more thoroughly described. (There is no need to make things look better/smoother, implementation is hard)
Some additional points
Abstract
*T2D should be mentioned in the title
*The abstract should mention the 300 initially contacted patients.
*Conclusions there is no need to start with the most positive data: patients were satisfied …. The methodology virtually guarantees that result (unhappy patients will drop out, not participate…)
Introduction
Nice overview of the literature but some details need work.
*The total annual cost of diabetes in de US is $327 BILION dollar …. (T1D + T2D)
http://care.diabetesjournals.org/content/early/2018/03/20/dci18-0007.full-text.pdf
*Line 36: ”Medication adherence is particularly important for patients with T2D as …” particularlythis suggest a contrast ..??? (T1D?)
*Line 51: “A benefit to pharmacists providing diabetes education is their proximity to medication dispensing. “ The patients are in need of benefits! …. (And those can than indirectly benefit pharmacists….)
* Line 51: maybe the authors want to comment on why only a minority of pharmacists have pursued the extra training
*Line 54: this sentence is not clear to me
Materials and Methods
*Line 75: More details are needed about the IRB (number, location, date) !
Did the patients received written details about the study?
Did they give their written consent?
Are their financial consequences for the patients?
*Line 82: the “3rd party web-based quality improvement and documentation platform” should be described in detail allowing replication of the study.
*Line 84: how was the selection of the 100-200 patients done?
*Line 156: the authors should elaborate on the use of “Reliability statistics”
Results
*Table 1: if the numbers between parenthesis are not % , what are they?
Line 171: Did all patients gave their consent concerning HbA1c levels?
Table 2 and Table 3 : maybe this data could be presented as in figure 3?
Figure 1 and 2
These are very hard to decipher and seem inappropriate for this kind of data.
A chart like in figure 4 seems more appropriate
It would be interesting to get a clear idea how many diabetes education classes were actually organized with the number of participants. We also need information about the background prevalence of this kind of intervention. Were these classes something completely new? How experienced were the educators?
Line 196: incomplete sentence
Line 211: for an international audience some more background context about the reimbursement of similar services would be interesting.
Discussion
Line 217: Some details about participation rates in other studies is probably needed
Line 210: Some power calculations should probably have been performed before the study in order to calibrate the sampling. This obviously is not possible after the study but based on the current knowledge an attempt could be made and discussed in this section guiding future researchers.
A discussion about the Hawthorn effect is needed and the effect size may be estimated from this study one again guiding future research.
Line 261: The collaborative aspect of the intervention needs to be more described in more detail: was it hard to obtain those prescriptions? Was there an attempt to give feedback to the physicians? When during the protocol were these prescriptions asked?
Line 275: details about “provider status legislation” should be explained (for an international audience)
Line 285: the raw data about limitations about the PDC should be in the result section
The discussion would certainly benefit from an economic analysis: what time-investment was needed? What where the total costs? What reimbursement is needed to make this intervention sustainable?
Conclusion
Line 293: explain what “typical billing mechanisms.” re
This section clearly needs much more work.
There is too much contrast with the conclusion in the abstract
What are the conclusions concerning the implementation process?
What will be done differently the next time?
What have the researchers learned?
How would they advise the Iowa Pharmacy Foundation about future projects?
....
Reviewer 3 Report
This manuscript uses passive verb tense numerous times that the reviewer was not able to count all the offenses. (e.g. Data was collected... patients were educated...etc.) Consider revising to active verb tense.
Line 60 need comma "pharmacist, this is called..."
Line 63-65 I would Omit this statement. I do not consider this to be a rationale but more the purpose. This is better explained in the objective paragraph starting line 66.
Line 68 What kind of comparisons were made? There needs to be some comparison analysis in the statistical analysis section. Table 1.
Line 76 Participants. There should be some sort of CONSORT chart to show where the participants were collected or excluded.
Line 101 "initial assessment and got received permission from the patient"
Line 109 and 117 What do you mean by "fit for synchronization"?
Line 120 need to change pronoun "...especially if the patient claimed they he or she do did not.."
Line 121 Need to elaborate on statement "Call allowed for answering questions"
Line 168 Table 1 It would be beneficial to test the differences in the 3 study groups.
Line 173 I cannot find any testing that show there was a benefit to PDCs in the synchronization group.
Table 2 and 3 could use comparison analysis
Line 179 I need better explanation on Figure 2 and 3 because there are numerous lines that trend downward that would explain a decrease in positive beliefs.
Line 186-187 Is there any discussion on why this patient had a negative effect?
Line 220 This would be beneficial to test the comparison to see if there was an extra benefit to the combined intervention that synchronization alone. This was not tested hence why it was difficult.
Line 279 Where was the randomization? Each participant elected which group to enroll in based on interest in each intervention.
Round 2
Reviewer 2 Report
There are some small things that still need some attention.
Overall responses are satisfactory
*The total cost of diabetes is still the wrong figure by a factor of 1000
*The IRB data is still missing (date-number)
*the answer to this comment is too vague "*Line 82: the “3rd party web-based quality improvement and documentation platform” should be described in detail allowing replication of the study" and this paragraph has not been carefully edited.
Author Response
Thank you for looking at the paper again.

Reviewer 3 Report
Thank you for your revision
Author Response

(The authors gave the same response as above.)
